# Adaptive Multi-Sensor Joint Tracking Algorithm with Unknown Noise Characteristics

**DOI:** 10.3390/s24113314

**Published:** 2024-05-22

**Authors:** Weihao Sun, Yi Wang, Weifeng Diao, Lin Zhou

**Affiliations:** Nanjing Research Institute of Electronics Technology, Nanjing 210039, China; sunwithheat@gmail.com (W.S.); wangyi15@nudt.edu.cn (Y.W.); diao_weifeng@163.com (W.D.)

**Keywords:** space-based optical sensors, orbital targets, target tracking, extended Kalman filter, self-adaptive methods

## Abstract

In this study, to solve the low accuracy of multi-space-based sensor joint tracking in the presence of unknown noise characteristics, an adaptive multi-sensor joint tracking algorithm (AMSJTA) is proposed. First, the coordinate transformation from the target object to the optical sensors is considered, and the observation vector-based measurement model is established. Then, the measurement noise characteristics are assumed to be white Gaussian noise, and the measurement covariance matrix is set as a constant. On this premise, the traditional iterative extended Kalman filter is applied to solve this problem. However, in most actual engineering applications, the measurement noise characteristics are unknown. Thus, a forgetting factor is introduced to adaptively estimate the unknown measurement noise characteristics, and the AMSJTA is designed to improve the tracking accuracy. Furthermore, the lower bound of the proposed algorithm is theoretically proved. Finally, numerical simulations are executed to verify the effectiveness and superiority of the proposed AMSJTA.

## 1. Introduction

The monitoring and tracking technology of near-Earth objects is currently one of the research hotspots in the field of aerospace. As the prerequisite and foundation of planetary defense, it has been a topic of concern that has received extensive attention from major institutions and scholars in various related fields [1,2,3]. Currently, the increasing prevalence of missile technology as a primary method of aerospace operation has led to numerous countries attaining mastery in this field, establishing it as an independent and effective combat approach. Consequently, the development of anti-missile defense technology and enhancement of missile defense capabilities have become crucial requirements for strategic defense systems. The primary function of a space-based early warning satellite system is to detect incoming threats (mainly ballistic missiles) and provide timely warnings and real-time detection data. A ballistic missile is a type of aircraft that utilizes ballistic control to follow predetermined flight paths. During their weightless free-flight phase, which is also the mid-segment of the flight process, warheads traverse predetermined orbits. Therefore, this segment also serves as a pivotal research area for orbit tracking and determination within early warning systems [4].

At present, the principal detection methods for near-Earth objects (NEOs) mainly include ground-based telescopes and space-based telescopes based on optical sensors and ground-based radars based on electromagnetic waves. As for ground-based radars, owing to the constraints of antenna aperture, power consumption, detection range, and susceptibility to atmospheric conditions and illumination, there might be difficulties, such as limited site selection, an insufficient observation time, a narrow detection area, and more constraints during the NEO process, which makes it difficult to accurately co-locate in complex environments, and a large number of optimization algorithms are required to determine the layout method for collaborative positioning [5,6,7]. In comparison, space-based optical sensors achieve real-time searching and tracking for NEOs, with the advantages of low energy consumption, high efficiency, wide area coverage, great measurement accuracy, long detection distances, etc. Moreover, the arrangement of space-based optical sensors is more flexible and less affected by the atmospheric environment [8]. These advantages make space-based optical detection a developing trend and the main direction for future NEO observation equipment. A constellation of several satellites also displays great advantages. Compared with a single spacecraft, a constellation has advantages with respect to capability, flexibility, and robustness. Space-based optical detection proves to be a crucial technique to attain space situational awareness. However, optical sensors can only obtain angle measurement information on NEOs and cannot directly obtain distance information. This passive angle positioning method cannot fully reflect the orbital information of the target. In order to obtain the three-dimensional position information of the target, at least two or more optical observation carriers are required to participate in localization. The progression of the cooperative orbit determination without the help of the ground station’s aid is one of the features of a satellite constellation. From the perspective of the task level, the basic tasks of space-based detection include various aspects, such as orbit maneuvering, attitude control, imaging models, etc. [9] Usually, comprehensive analytical assessments and numerical simulations of observability are required in the simulation process [10]. In this study, the filtering problems in the determination of a target’s trajectory are the main focus.

One of the major challenges in multi-space-based sensor joint tracking scenarios is the low accuracy due to the presence of unknown noise characteristics. In tracking applications, the use of the Cartesian coordinate system for the modeling process and the fact that the measurements are usually in a polar or spherical coordinate system, as well as the motion model of the space target, all lead to the issue that the target tracking process is essentially a nonlinear filtering problem [11]. Nonlinear filtering algorithms, such as the extended Kalman filter, unscented Kalman filter, and particle filter, can perform well in their respective applicable scenarios to improve measurement accuracy [12,13,14,15]. There are also many current studies that have used various types of improved filtering methods for radar measurements or sensor measurements of a target’s motion trajectory. For instance, the tracking problem of high-speed gliding target tracking with unknown time-varying maneuvers was investigated in [16], and a robust extended Kalman filter based on the compensation of maneuver observers was proposed to solve the model mismatch problem by using the output of maneuver observers to correct the prediction step of the dynamics. A method for initial trajectory determination was proposed in [17], which can be applied to space-based space target monitoring, solving the effective result and reducing the susceptibility to the initial value of distance in space-based space target monitoring. In [18], a method of quantized extended Kalman filtering under quantized measurement conditions was derived, and the measurement update was based on the conditional mean estimation of the given quantized measurements, which improves the accuracy of trajectory tracking under the condition of quantized noise. There are also methods that utilize artificial intelligence to assist filtering. For example, in [19], an information fusion technology based on artificial neural networks was proposed for multi-sensor integrated autonomous navigation. Most of the current methods have been able to realize high precision under Gaussian noise, but most of them require large amounts of feedback data. We hope to offer an available option to solve the problem of low accuracy in tracking in the presence of unknown noise characteristics when the data are not abundant. Furthermore, the tracking of target trajectories is also of great importance in the task allocation of space-based systems [20,21,22].

In this study, we propose an adaptive multi-sensor joint tracking algorithm (AMSJTA). First, the orbiting and tracking model of NEOs based on space-based optical observation is investigated. Then, the orbital dynamics model of NEOs and the observation model of optical sensors are established. According to the acquisition mode and characteristics of space-based optical goniometric information, the corresponding tracking process is provided. Noting that the values of the process noise covariance matrix Q and the measurement noise covariance matrix R are highly influential on the values of the extended Kalman filter [23], the adaptive method is introduced here to adaptively update the process noise and the measurement noise at each step of the filtering process, so as to improve the performance of the iterative extended Kalman filter and enhance the accuracy of the filtering. A simple derivation of the Bayesian CRLB for the estimation error is also carried out as an intuitive alternative to the estimation error to be used as a tracking performance metric for the target.

The remainder of this article is organized as follows: Section 2 presents the dynamic observation model of space-based multi-optical sensors for orbital targets. In Section 3, the AMSJTA filtering method is specified, and the related derivation procedure is provided. Section 4 presents a discussion and the simulation results of an application of the proposed algorithm in a specific scenario. Finally, Section 5 presents the conclusions.

## 2. Target Motion and Measurement Model

The models in this study are mainly implemented in the Earth-centered, Earth-fixed coordinate system (ECEF), the Earth-centered inertial coordinate system (ECI), and the coordinate system based on the sensor itself, which are described as follows:The ECEF coordinate system is a typical Cartesian coordinate system, which is fixed relative to the Earth. The origin of the coordinate system O is located in the center of the Earth, the OX axis is in the equatorial plane and points to the meridian where the Greenwich Observatory is located, the OZ axis is perpendicular to the equatorial plane and points toward the North Pole direction when combined with the axis of the rotation of the Earth, and the relationship between the OX, OY, and OZ axes satisfies the right-hand rule.The ECI coordinate system is also a typical Cartesian coordinate system, which is stationary relative to the fixed star. It is usually considered to be in the inertial space, with the geocentric point O as the coordinate origin. OX is in the equatorial plane pointing to the equinox point. OZ is perpendicular to the equatorial plane and points to the North Pole, and the orientation of the OY axis conforms to the right-hand rule.In the orbital coordinate system in which the coordinate origin O of the sensor body coordinate system is the geometric center of the sensor, the OX axis is the extension of the line connecting the Earth’s center and the coordinate origin, the OZ axis is located in the plane formed by the OX axis and the Earth’s rotation axis and is perpendicular to the OX axis, and the orientation of the OY axis satisfies the right-hand rule.

According to the principle of coordinate rotation transformation, the transfer matrix from the orbital coordinate system to the ECI coordinate system is as follows:(1)$$\left(\begin{array}{ccc} cos\omega & -sin\omega & 0\\ sin\omega & cos\omega & 0\\ 0 & 0 & 1 \end{array}\right)\cdot \left(\begin{array}{ccc} 1 & 0 & 0\\ 0 & cosi & -sini\\ 0 & sini & cosi \end{array}\right)\cdot \left(\begin{array}{ccc} cos\Omega & -sin\Omega & 0\\ sin\Omega & cos\Omega & 0\\ 0 & 0 & 1 \end{array}\right)$$
where ω is the argument of periapsis, i is the inclination, and Ω is the longitude of the ascending node.

Firstly, the basic structure and function of the space-based multi-sensor system are analyzed and studied, the implementation process of the space-based multi-sensor early warning and detection mission along with the working mode and detection capability of the space-based multi-sensor are discussed, and the working model in the middle part of the ballistic trajectory tracking of multiple targets is established. 

When solving the observation vector, it is necessary to obtain the position information of the observation satellite. The position vector of the observation satellite is first calculated according to the orbital mechanics. Considering that the motion of the LEO satellite around the Earth can be approximated as only being acted upon by the gravitational force of the Earth, if the Earth is equivalent to a mass point, the Earth and the satellite constitute a simple two-body system as shown in Figure 1. The orbits can be constructed by using the orbital elements for orbit reconstruction [24]. In this study, the method of dynamical equations is adopted to establish the continuous state equation of the space target. The fourth-order Runge–Kutta method is applied to obtain the nonlinear discrete equation of the space target [25].

In the ECEF coordinate system, it is assumed that the position and velocity of space targets are expressed as follows:(2)$$\left\{\begin{array}{c} \vec{r}={\left(x,y,z\right)}^{T}\;\\ \;\vec{v}={\;(v_{x},v_{y},v_{z})}^{T} \end{array}\right.$$

Considering the effect of the J2 perturbation term, the orbital dynamics equations for the space target are as follows:(3)$$\frac{d\vec{r}}{dt}=\vec{v}\;\frac{d\vec{v}}{dt}=\left[\begin{array}{c} -\frac{\mu_{e}}{r^{3}}(1+\frac{c_{e}}{r^{2}}(1-5{\left(\frac{z}{r}\right)}^{2}))\cdot x+{\omega_{e}}^{2}\cdot x+2\omega_{e}\cdot v_{y}\\ -\frac{\mu_{e}}{r^{3}}(1+\frac{c_{e}}{r^{2}}(1-5{\left(\frac{z}{r}\right)}^{2}))\cdot y+{\omega_{e}}^{2}\cdot y-2\omega_{e}\cdot v_{x}\\ -\frac{\mu_{e}}{r^{3}}(1+\frac{c_{e}}{r^{2}}(1-5{\left(\frac{z}{r}\right)}^{2}))\cdot z \end{array}\right]$$
where $c_{e}=\frac{3J_{2}{R_{2}}^{2}}{2}$. According to these formulas, the continuous equation of the state of the space target is expanded by the fourth-order Runge–Kutta method. The acceleration can be obtained according to Equation (2). Then, the nonlinear discrete equation of the state of the space target can be expressed as follows:(4)$$\vec{X_{k+1}}=f(\vec{X_{k}})+\varepsilon_{k}$$

The process noise covariance matrix is set as $Q=E(\varepsilon_{k}{\varepsilon_{k}}^{T})$. According to [26], it can be described using the Singer model in the following form:(5)$$Q=\left[\begin{array}{ccc} \frac{T^{5}}{20} & \frac{T^{4}}{8} & \frac{T^{3}}{6}\\ \frac{T^{4}}{8} & \frac{T^{3}}{3} & \frac{T^{2}}{2}\\ \frac{T^{3}}{6} & \frac{T^{2}}{2} & T \end{array}\right]\times \sigma^{2}$$

For the signals acquired by the sensors, a spatial polar coordinate system is applied for processing as shown in Figure 2. The measurement vectors of the observed target in the coordinate system are defined using the direction angle and the pitch angle as follows:(6)$$\left(\begin{array}{c} \theta_{i}\\ \varphi_{i} \end{array}\right)=h(r_{s})=\left(\begin{array}{c} arccos(\frac{z_{T}}{\sqrt{{y_{T}}^{2}+{z_{T}}^{2}}})\\ -arcsin(\frac{x_{T}}{\sqrt{{{{x_{T}}^{2}+y}_{T}}^{2}+{z_{T}}^{2}}}) \end{array}\right)$$

For the optical sensors, they can only acquire the angle information of the target. As a result, in order to determine the spatial location of the target, we need the observation results of at least two sensors and their spatial location data. In the case of dual-satellite observation and orbit determination, measurement errors are considered as follows:(7)$$\vec{Z}=g(\vec{X})+v=g(r_{1},\;\theta_{1},\;\varphi_{1},\;r_{2},\;\theta_{2},\;\varphi_{2})+v$$

Using the observed angle information of at least two sensors, along with the spatial information of the sensors themselves, we have enough information to obtain the spatial information of the observation target. Therefore, the observation can be determined as the spatial position coordinates of the target constructed from the direction and pitch angles obtained from at least two telescope satellites. The observation error is defined as:(8)$$R=E(vv^{T})$$

For optical sensors, since they can only obtain the angle information during observation, we can assume that there is a positioning ray that starts from the center of the optical sensor and points in the direction of the target. Under ideal conditions, the positioning rays of multiple satellites for a single observation target should intersect in space. However, due to the influence of various systematic errors and random errors, it is very likely that these rays from multiple satellites will have no common intersection point in space under practical conditions. Using a system of linear equations to directly solve the tracking problem may often result in the inability to yield the final outcomes. In the case of two satellites cooperating, the midpoint of the common perpendicular between two positioning rays in space can be used instead of the point of intersection for the operation as shown in Figure 3. A and B are the intersection points of the positioning rays from two satellites and the common perpendicular. However, in the case of more than two satellites observing simultaneously, Newton’s method can be used to iteratively obtain the point in space that minimizes the sum of the distances from each ray [27].

It is noted that since the final results of the extended Kalman filter are highly influenced by the values of the process noise covariance matrix Q and the measurement noise covariance matrix R, improper values of Q and R can greatly reduce the accuracy of the filter results and even lead to filter dispersion. In most studies of dynamic state prediction, the values of Q and R are set as fixed constants during the estimation process, and it is often difficult to obtain sufficiently good results for dynamic processes by using the exhaustive method. Therefore, the adaptive estimation method used in this study is to adaptively adjust the measurement noise covariance matrix R and the process noise covariance matrix Q. Then, R and Q are updated according to the current state values at each step of the iterative extended Kalman filtering so as to further improve the filtering results.

## 3. Adaptive Multi-Sensor Joint Tracking Algorithm

Since the observation equations are nonlinear, extended Kalman filtering is required for the state estimation of the target. According to the established motion equations and observation equations, the traditional extended Kalman filtering process is as follows in Algorithm 1. The status transition matrix F in extended Kalman filtering is different from the case in linear filtering and can be replaced by the Jacobian matrix. The Taylor series expansion can be retained as the third- or fourth-order terms. However, the second-order term is most usually applied for the comprehensive consideration of both computational complexity and performance.
**Algorithm 1**: EKF$1.\;\mathrm{Inputs}:\mathrm{initial}\;\mathrm{estimates}\;{\hat{X}}_{0},\;P_{0}$2. Set *k* = 03. Repeat:  *k* = *k* + 1Prediction process of extended Kalman filtering:                ${\hat{X}}_{k+1|k}=f({\hat{X}}_{k|k})$              $P_{k+1|k}=FP_{k}F^{T}+Q$
Kalman gain update:             
$K=P_{k+1|k}F^{T}{\left(FP_{k+1|k}F^{T}\right)}^{-1}$
 Residuals of predictions:             
vk=Zk+1−Hk+1X^k+1|k
Update process of extended Kalman filtering:               Xk=Xk+Kvk            
Pk+1|k+1=(I−KHk+1)Pk+1|k
  Exit the loop when there is no new measured state quantity4. Output:Xk


At the same time, an iterative approach is applied, where the results of the k^th^ moments that have been obtained are processed iteratively again, and the outcome after each iteration is used as the input for the next iteration so that a better performance can be achieved for the filtering results for each state $X_{k}$.

When the data available are limited, we can apply the least squares method for the determination of the initial orbit. The original trail can be determined through second-order polynomial fitting using the initial data. We can define the residual as the difference between the actual observation and the estimated observation. With the idea of the least squares method, we can minimize the sum of squared residuals of the system within the observation interval. During the iterative process, the covariance can be written as follows:(9)$$f={\left(X_{k+1}-X_{k+1|k}\right)}^{T}P^{-1}(X_{k+1}-X_{k+1|k})+{\left(h(X)-Z_{k+1}\right)}^{T}R^{-1}(h(X)-Z_{k+1})$$

After each iteration, the covariance proceeds in a decreasing trend. For each iteration, the update of $X_{k}$ is:(10)$$∆X={\left({P_{k+1|k}}^{-1}+H^{T}R^{-1}H\right)}^{-1}({P_{k+1|k}}^{-1}(X_{k+1}-X_{k+1|k})\;+H^{T}R^{-1}(h({\hat{X}}_{k+1|k})-Z_{k+1}))\;$$

The iteration is terminated when the reduction in the covariance between two iterations falls below a threshold.

In this study, an adaptive filtering approach is adopted for further improvement in the effect of the EKF to obtain better optimization results, as shown in the following equation:(11)$$C_{v_{k}}=\frac{1}{N}\sum_{j=0}^{N}v_{k-j}{v_{k-1}}^{T}$$
where $v_{k}$ is the error vector at time *k*, which can be taken as the residual vector or the state error depending on the object to be adapted and can be used to perform the corresponding covariance matrix, $R_{k}$ or $Q_{k}$. According to [28], the following expressions can be obtained:(12)$$R_{k}=v_{k}{v_{k}}^{T}+H_{k}{\hat{P}}_{k}{H_{k}}^{T}=C_{v_{k}}+H_{k}{\hat{P}}_{k}{H_{k}}^{T}\;$$
(13)$$Q_{k}=\frac{1}{N}\sum_{j=j_{0}}^{k}∆x_{j}{∆x_{j}}^{T}+{\hat{P}}_{k}-\;F_{k}{\hat{P}}_{k-1}{F_{k}}^{T}\;\;$$

The traditional extended Kalman filter algorithm has difficulties in accurately determining the system process noise and the measurement noise. Some existing adaptive methods have deficiencies in estimating the covariance of measurement noise while updating the state. Therefore, in order to cope with this challenge, this study proposes an improved adaptive method to determine $R_{k}$ and $Q_{k}$ in the EKF process. The covariance matching method can adjust its covariance matrix according to the theoretical value of innovation. In the prediction step of the EKF, the innovation is the difference between the actual measured value and its predicted value, which can be calculated as follows:(14)$$v_{k}=Z_{k+1}-H_{k+1}{\hat{X}}_{k+1|k}$$

Compared with some methods that directly use the residuals estimated at the current moment, here, a sliding window is used for the $\varepsilon_{k}{\varepsilon_{k}}^{T}$ approximation by taking the average value as follows:(15)$$\Phi_{k}=\frac{1}{M}\sum_{i=k-M+1}^{k}\varepsilon_{i}{\varepsilon_{i}}^{T}$$

Meanwhile, this study introduces the forgetting factor α to adaptively estimate $R_{k}$, which satisfies the following equation:(16)$$\alpha =\frac{1-d}{1-d^{k}}$$

A smaller α gives more weight to the previous estimate and, therefore, prevents excessive fluctuations in the adaptive process and longer time delays in order to adapt to changes. According to the principle of automated control, *d* is taken as 0.95.
(17)$$R_{k}=(1-\alpha )R_{k-1}+\alpha (\Phi_{k}+H_{k}P{H_{k}}^{T})\;$$
(18)$$Q_{k}=(1-\alpha )Q_{k-1}+\alpha (\Phi_{k}+{\hat{P}}_{k}-\;F_{k}{\hat{P}}_{k-1}{F_{k}}^{T})\;\;$$

A brief representation of the AMSJTA process in this study is shown below in Figure 4:



(19)
$${\tilde{x}}_{k+1|k}=x_{k+1}-\;{\hat{x}}_{k+1|k}$$



Due to the one-step prediction error of the state, the following expressions can be obtained.

The filtering residuals:(20)$$\gamma_{k+1}=z_{k+1}-\;{\hat{z}}_{k+1|k}$$

The state estimation error:(21)$${\tilde{x}}_{k+1|k+1}=x_{k+1}-\;{\hat{x}}_{k+1|k+1}$$

A series of separately defined positive definite matrices exist as follows:(22)$$\left\{\begin{array}{c} {\Phi_{k+1|k}}^{*}>\Phi_{k+1|k}≝\;E({\tilde{x}}_{k+1|k}{{\tilde{x}}_{k+1|k}}^{T})\\ {S_{k+1}}^{*}>S_{k+1}≝\;E(\gamma_{k+1}{\gamma_{k+1}}^{T})\\ {\Phi_{k+1|k+1}}^{*}>\Phi_{k+1|k+1}≝\;E({\tilde{x}}_{k+1|k+1}{{\tilde{x}}_{k+1|k+1}}^{T}) \end{array}\right.$$

According to [29], it is known that such a construction exists; therefore, the initial condition can be satisfied as follows:(23)$${\Phi_{0|0}}^{*}>\Phi_{0|0}\;{S_{0}}^{*}>S_{0}$$

Then, for the Kalman gain $K_{k+1}$ at time *k* − 1, the following expressions can be applied:(24)$${\Phi_{k+1|k+1}|}_{K_{k+1}}\leq {{\Phi^{*}}_{k+1|k+1}|}_{{\varepsilon^{opt}}_{k+1},\;{K^{opt}}_{k+1}}\;\leq {{\Phi^{*}}_{k+1|k+1}|}_{\varepsilon_{k+1},\;K_{k+1}}\;{\;S}_{k+1}\leq {{S_{k+1}}^{*}|}_{{\varepsilon^{opt}}_{k+1}}\;\leq {{S_{k+1}}^{*}|}_{\varepsilon_{k+1}}\;{K^{opt}}_{k+1}={\Phi_{k+1|k}}^{*}{H_{k+1}}^{T}{\left({S_{k+1}}^{*}\right)}^{-1}$$

Next, pseudo-system state transfer matrices and measurement matrices are constructed using statistical linear error propagation methods:(25)$${\tilde{H}}_{k+1}≜{P_{k+1|k}}^{T}I_{{\hat{X}}_{k|k-1}}\;{\tilde{F}}_{k}≜{P_{k|k}}^{T}I_{X_{k}}$$

Then, the original state transfer equation and measurement equation can be linearized as follows:(26)$$x_{k}=\alpha_{k-1}\;{\tilde{F}}_{k-1}x_{k-1}+\omega_{k-1}$$
(27)$$z_{k}=\beta_{k}\;{\tilde{H}}_{k}x_{k}+v_{k}$$
where $\alpha_{k}$ and $\beta_{k}$ are the compensating diagonal matrices used to minimize possible estimation errors. The one-step prediction of the information matrix is as follows:(28)$$I_{{\hat{X}}_{k|k-1}}={\left[(\alpha_{k-1}\;{\tilde{F}}_{k-1}){I_{k-1|k-1}}^{-1}{\left(\alpha_{k-1}\;{\tilde{F}}_{k-1}\right)}^{T}+Q_{k-1}\right]}^{-1}\;$$

Considering the nonlinear stochastic system presented earlier and the adaptive iterative extended Kalman filtering algorithm described in the previous section, the error is exponentially bounded on the mean square under these assumptions [30].

Finally, a brief discussion of the existence of error bounds for adaptive filtering is presented. Consider the discrete nonlinear system as described below:(29)$$X_{k+1}=f_{k}(X_{k},v_{k})\;z_{k+1}=h_{k+1}(X_{k+1},w_{k+1})$$

Since $v_{k}$ and $w_{k}$ are white noise independent of each other here and independent of the initial state density function of the system, assume that the set of measurements up to moment k is denoted as $Z_{k}={\left\{z_{i}\right\}}_{i=1}^{k}$, the unbiased estimate of $X_{k}$ is denoted as ${\hat{X}}_{k|k}$, and the covariance matrix is denoted as $P_{k|k}$. The lower bound of this matrix is denoted as CRLB and satisfies the following equation:(30)$$P_{k|k}=E[({\hat{X}}_{k}(P_{k})-X_{k}){\left({\hat{X}}_{k}(P_{k})-X_{k}\right)}^{T}]\geq {\left[I_{X_{k}}(P_{k})\right]}^{-1}$$
where $I_{k}$ is the Fisher information matrix (FIM).
(31)$$I_{X_{k}}(P_{k})=E\{[\nabla_{X_{k}}lnp(X_{k},Z_{k})]{\left[\nabla_{X_{k}}lnp(X_{k},Z_{k})\right]}^{T}\}$$

For the above Gaussian model, the FIM can be written as the sum of the following two components:(32)$$I_{X_{k}}(P_{k})\;≜I_{k}+I_{Z}(P_{k},x_{k})$$
where $I_{k}$ corresponds to the a priori information about the state of the target, and $I_{Z}(P_{k},x_{k})$ is the part generated during the measurement process.
(33)$$I_{k}={\left[Q_{k}+FI_{X_{k-1}}(P_{k-1})F^{T}\right]}^{-1}\;I_{Z}(P_{k},x_{k})=E\{\nabla_{X_{k}}^{T}h(x_{k}){\left[R_{k}(P_{k})\right]}^{-1}\nabla_{X_{k}}h(x_{k})\}$$

The calculations can be simplified by utilizing predicted values instead of statistical averages, as given by the following equation:(34)$$I_{{\hat{X}}_{k|k-1}}(P_{k})=I_{k}+H_{k}^{T}{\left[R_{k}(P_{k})\right]}^{-1}H_{k}$$

## 4. Simulation and Validation

In this section, we present simulation experimental scenarios and results to demonstrate that the improved adaptive iterative extended Kalman filter has better performance in applications. Moreover, in addition to the comparison in terms of performance, we also investigate the possible effects of changes in the initial conditions on the filtering results. A simulation scenario of the joint observation of multiple space-based optical transmitters for a low-orbit target is constructed to validate the performance of the AMSJTA in this study. The parameters used in the simulation are listed in the following Table 1.

For the simulation of the orbit of the near-Earth satellites, both satellites carry space-based optical sensors. According to Section 2, the method of the four-stage Runge–Kutta expansion under the consideration of the influence of the Earth’s *J*_2_ perturbation term is adopted. The initial state of the two satellites and the space target is shown in the following Table 2 in ECEF coordinates.

The trajectories of the satellites and a space target in the ECEF coordinate system are shown in the following Figure 5.

The measurement period of each of the two observation satellites for the target is dt = 1 s. The standard deviation and covariance matrices of the sensor’s random measurement noise are as follows:$$\sigma_{r}=1000\;m,\sigma_{a}=0.1{}^{\circ},\sigma_{e}=0.1{}^{\circ}$$
$$R_{0}=diag([\sigma_{r},\sigma_{a},\sigma_{e}])$$

The sensors of each satellite need to acquire angular degrees in their respective coordinate systems and jointly determine the orbit in the ECI coordinate system, the true trajectory in the ECI coordinate system, the observation trajectory, and the trajectory after adaptive iterative extended Kalman filtering.

The root-mean-square error (RMSE) of the target position can be used to evaluate the performance of the filtering effect. The tracking results of the iterative extended Kalman filter and the filter with the addition of adaptive tuning are compared, as shown in Figure 6. In the results in the figure, it can be observed that after adding the adaptive process, the method improves the tracking accuracy. Therefore, the designed AMSJTA has a better performance compared with the conventional IEKF filter, and the filtered error is basically maintained at around 0.1%.

In order to better demonstrate the advantages of the adaptive method, the initial $Q_{k}$ values are artificially biased to observe the performance of the traditional filtering method and the filtering method with adaptive improvement. The results of 1000 Monte Carlo simulations are taken in the experiment.

First, take $Q=10Q_{0}$. In this case, according to Figure 7, it can be clearly observed that the conventional IEKF process estimation bias is significantly improved, while the use of adaptive methods can still maintain a relatively acceptable filtering performance, greatly enhancing the stability of the original algorithm.

In another case, instead of Gaussian white noise, the inserted system noise Q is made to be a random noise that gradually decreases with time, oriented as shown in Figure 8. In this example, the adaptive processing of the filtering can still converge to a suitable accuracy relatively quickly, but the error of the conventional filtering method becomes larger and even appears to have a possible tendency toward divergence.

The results obtained from the simulation experiments are basically consistent with the previous conclusions. The results of this study demonstrate that the proposed adaptive algorithm can obtain a better filtering performance and improve the robustness of the extended Kalman filtering algorithm.

## 5. Conclusions

As the most concise and direct method for nonlinear problems such as orbital target tracking, the extended Kalman filter is deficient in stability. The proposed AMSJTA method compensates for the shortcomings of the EKF’s instability and significantly improves the accuracy of tracking orbital targets. The improved algorithm combines the least-squares criterion and adaptive iterative extended Kalman filtering for orbit determination. Based on the covariance matching theory, this study established and utilized the known residual quantities of the filtering process to update the measurement state noise and the process noise covariance. The proposed adaptive algorithm improves the accuracy of the estimation of target orbits and achieves better performance than traditional extended Kalman filtering with unknown measurement noise characteristics.

## Figures and Tables

**Figure 1 sensors-24-03314-f001:**
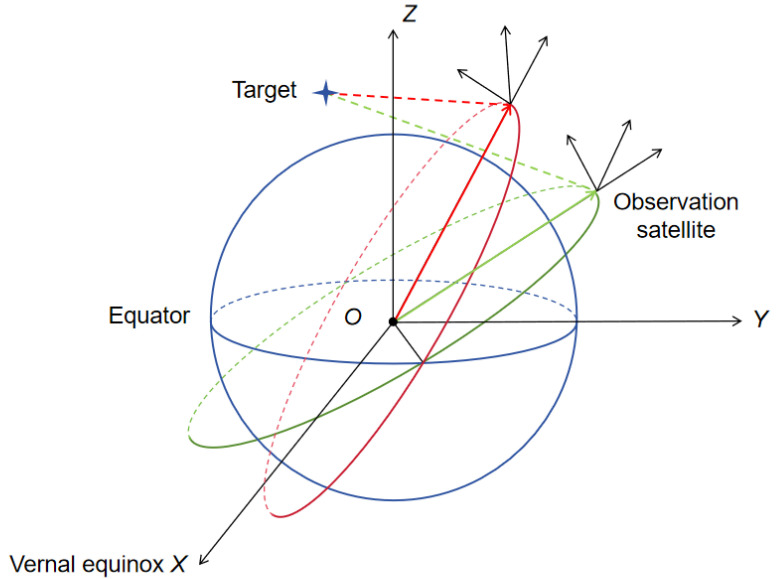
A geometric diagram of multiple space-based optical sensors for orbital target observations.

**Figure 2 sensors-24-03314-f002:**
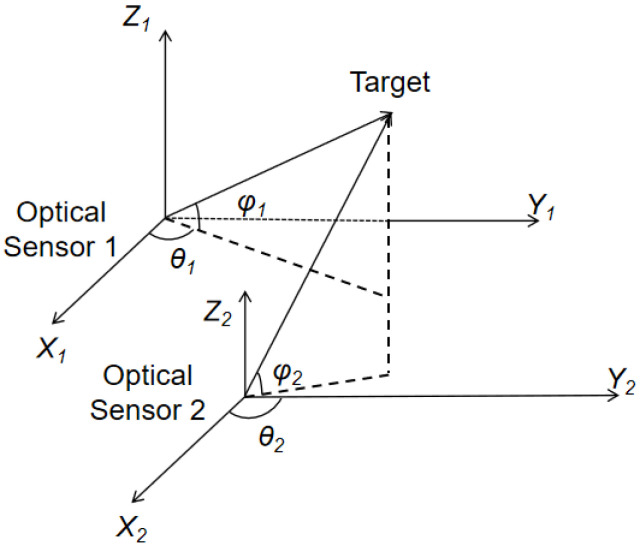
Diagram of polar coordinate observation of a target in the sensor body coordinate system.

**Figure 3 sensors-24-03314-f003:**
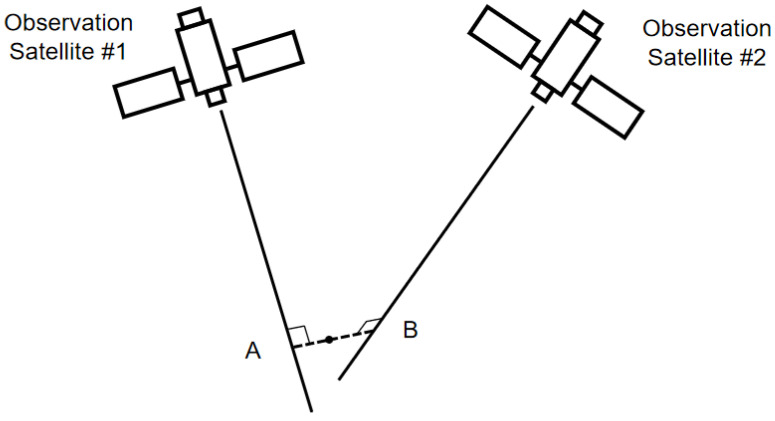
Positioning rays without intersections.

**Figure 4 sensors-24-03314-f004:**
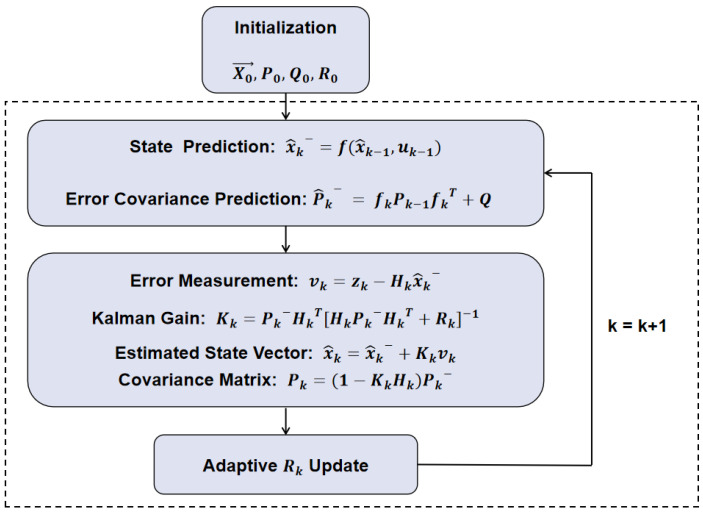
Diagram of the adaptive multi-sensor joint tracking algorithm process.

**Figure 5 sensors-24-03314-f005:**
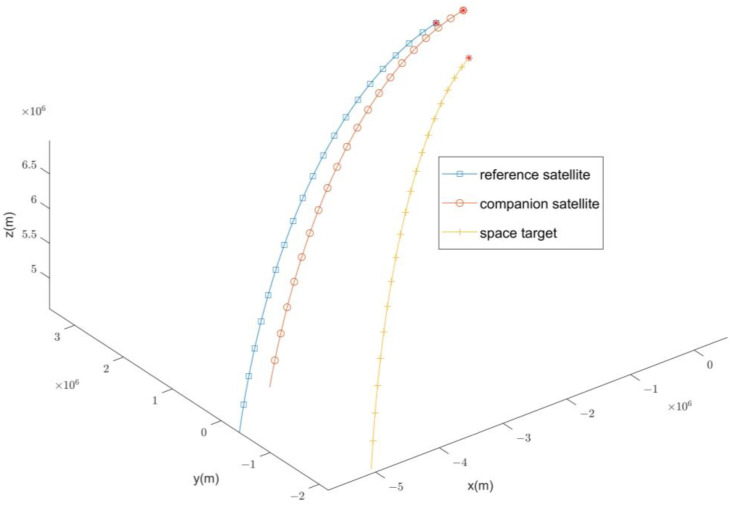
Satellite and space target trajectories.

**Figure 6 sensors-24-03314-f006:**
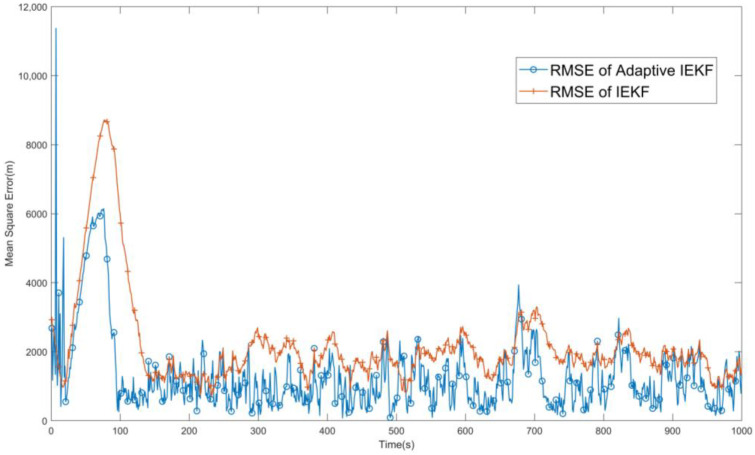
Comparison of RMSE after IEKF and AMSJTA filtering treatment.

**Figure 7 sensors-24-03314-f007:**
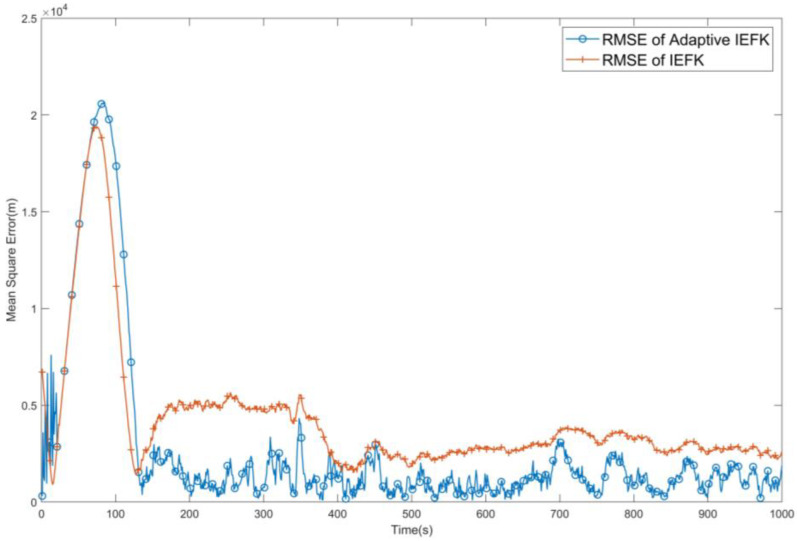
Comparison of RMSE after IEKF and AMSJTA filtering when $Q=10Q_{0}$.

**Figure 8 sensors-24-03314-f008:**
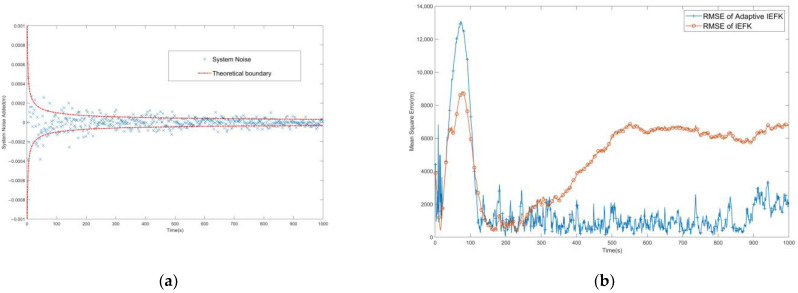
(**a**) Diagram of the inserted system noise Q change over time. (**b**) Comparison of RMSE after IEKF and AMSJTA filtering when Q decreases over time.

**Table 1 sensors-24-03314-t001:** Simulation parameters.

Parameters	Notation	Value
Equatorial radius of the Earth	$R_{e}$	6.37814 × 10^6^ (m)
Earth’s gravitational constant	$\mu_{e}$	3.986006 × 10^14^ (m^3^/s)^2^
Angular velocity of the Earth’s rotation	$\omega_{e}$	7.292115 × 10^−5^ (rad/s)
*J*_2_ constant	*J* _2_	1.082626836 × 10^−3^

**Table 2 sensors-24-03314-t002:** Initial state vectors for space units.

Unit (of Measure)	x/m	y/m	z/m
Reference satellite (Sensor #1)	$3.23\times 10^{5}$	$3.51\times 10^{6}$	$6.54\times 10^{6}$
Companion satellite (Sensor #2)	$5.18\times 10^{5}$	$3.21\times 10^{6}$	$6.79\times 10^{6}$
Space target	$2.30\times 10^{-9}$	$2.42\times 10^{6}$	$6.65\times 10^{6}$
Unit (of measure)	$v_{x}/(m\cdot s^{-1})$	$v_{y}/(m\cdot s^{-1})$	$v_{z}/(m\cdot s^{-1})$
Reference satellite (Sensor #1)	$-6.78\times 10^{3}$	$-3.20\times 10^{3}$	$1.30\times 10^{3}$
Companion satellite (Sensor #2)	$-6.35\times 10^{3}$	$-3.11\times 10^{3}$	$1.56\times 10^{3}$
Space target	$-5.76\times 10^{3}$	$-4.54\times 10^{3}$	$1.64\times 10^{3}$

## Data Availability

The data are contained within this article.

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
