# Peer review of "Adaptive Multi-Sensor Joint Tracking Algorithm with Unknown Noise Characteristics"

_sensors, 2024, doi:10.3390/s24113314_

Round 1
Reviewer 1 Report
Comments and Suggestions for Authors
This paper an adaptive multi-sensor joint tracking algorithm (AMSJTA) to solve the low accuracy of multi-spaced-based-sensor joint tracking in the presence of unknown noise characteristics.
The following comments must be addressed before publication.
1. lines 98-102: chapter 1,2,3,IV,5, two different types of numbers are employed.
2. line 141: citation should be added for the RK method:
for example, J. R. Dormand, P. J. Prince, A family of embedded Runge-Kutta formulae, Journal of Computational and Applied Mathematics 6 (1) (1980) 19(26).
3. Algorithm 1: both bold and unbold variables are employed: F
4. line 248: since v_k and w_k: lacks a space between the "and" and "w_k"
5. line 245: citation [26] should be placed before the full stop.
6. lines 218-219: an equation is placed between two lines; correct it
7. line 212; lacks a space before R_k
8. Contribution against Ref. [24] should be highlighted
9. After line 84, a summary of the current methods, including the advantages and disadvantages, as well as your motivation, should be added.
10. It is recommended to add the following references:
Zhou, X., Qin, T., Macdonald, M., and Qiao, D. “Observability Analysis of Cooperative Orbit Determination Using Inertial Inter-Spacecraft Angle Measurements.” Acta Astronautica, Vol. 210, 2023, pp. 289–302. https://doi.org/10.1016/j.actaastro.2023.05.019.
Yuan, Y., Yu, F., and Zong, H. “Multisensor Integrated Autonomous Navigation Based on Intelligent Information Fusion.” Journal of Spacecraft and Rockets, 2024, pp. 1–11. https://doi.org/10.2514/1.A35585.
Comments on the Quality of English Languageminor revision
Reviewer 2 Report
Comments and Suggestions for Authors
Please see my comments in the attached file.

the English is awkward. The paper needs a strong edit.
Round 2
Reviewer 1 Report
Comments and Suggestions for Authors
I have no more comments. It is fine.
Author Response
Dear reviewer:
On behalf of all my co-authors, we are very grateful for your kind approval. Thank you for your time and dedication in reviewing our manuscript. We are honored to have the opportunity to receive comments and suggestions from experts like you. The reviewing and communication experience during this period has greatly benefited us as well.
Once again we would like to express our great appreciation to you and your comments. Wish you a pleasant day.
Best wishes
Reviewer 2 Report
Comments and Suggestions for Authors
I have been through the revised manuscript. The authors have worked hard to improve the manuscript.
Please clarify further what the observation is. Two options:- The observation is the direction angle and pitch angle referenced in eq 2.6
- The observation is the position coordinates of the target (constructed deterministically from the direction and pitch angles from two or more telescope satellites)
- The initial orbit determination (IOD) process where minimal data is available
- The orbit refinement process when lots of data is available (eq 2.6 is the obs model)
Author Response
Dear reviewer:
On behalf of all my co-authors, we are very grateful for your kind suggestions. Thank you for your time and dedication in reviewing our manuscript. We are honored to have the opportunity to receive comments and suggestions from experts like you. The reviewing and communication experience during this period has greatly benefited us as well. Further revisions are marked yellow in the newly updated version.
We have further clarify the observation according to option 2 as the position coordinates of the target determined from the direction and pitch angles from at least 2 satellites.
We greatly appreciate the orbit determination problems that you mentioned and they gave us some inspirations as well. We have given some discussions on the idea of least squares method in the IOD problems and hopefully they can improve the manuscript.
For the refinement process with large amount of data, it is exactly within our subsequent research plan considering the scheduling problem. We will focus more on this issue considering that orbit refinement is essentially data smoothing process for the purpose of more accurate prediction. We greatly appreciate your research idea that provided us with an overall perspective and refresh our minds.
Once again we would like to express our great appreciation to you and your comments. Wish you a pleasant day.
Best wishes